# Digital Healthcare Innovative Services in Times of Crisis: A Literature Review

**DOI:** 10.3390/healthcare13080889

**Published:** 2025-04-12

**Authors:** Olympia Anastasiadou, Markos Tsipouras, Panagiotis Mpogiatzidis, Pantelis Angelidis

**Affiliations:** 1Nursing Department, General Hospital of G. Gennimatas, 54635 Thessaloniki, Greece; 2Department of Electrical and Computer Engineering, University of Western Macedonia, Ypatias 59 Anw Touba, 54351 Thessaloniki, Greece; mtsipouras@uowm.gr; 3Department of Midwifery, School of Health Sciences, University of Western Macedonia, 54635 Thessaloniki, Greece; bogiatzidis@yahoo.com; 44th Healthcare Authority of Greece, 54695 Thessaloniki, Greece; 5Biomedical Technology and Digital Health Laboratory, Department of Electrical and Computer Engineering, University of Western Macedonia, 54635 Thessaloniki, Greece; paggelidis@uowm.gr

**Keywords:** digital health, digital technologies, management implications, digital platforms

## Abstract

**Objectives:** The transformation of healthcare systems during crises, particularly demonstrated during the COVID-19 pandemic, emphasizes the urgent need for effective research methodologies to evaluate digital healthcare innovations. These methodologies are essential in addressing the rapid shift in healthcare service delivery modalities, responding to unprecedented challenges that have revealed both opportunities and barriers within the digital ecosystem. **Methods:** For this review, research was carried out on the Medline/PubMed, Scopus, and Google Scholar databases to locate articles published from 2015 to 2024, using the search terms digital health, digital technologies, management implications, and digital platforms. The inclusion criteria referred to studies that were directly related to the topic, available in the English language, and published in peer-reviewed scientific journals. The exclusion parameters were as follows: (a) articles not relevant to the topic as defined in the purpose of the review, (b) systematic reviews and meta-analyses, and (c) articles published in a language other than English. **Results:** Key findings indicate that, while digital health technologies have the potential to mitigate healthcare disparities, they often exacerbate existing inequities, especially among vulnerable populations lacking consistent access to technology. Furthermore, the shift towards digital platforms has revealed significant gaps in workforce training and support, which are essential for effective implementation. **Conclusions:** This review underscores the financial implications, with expenditures rising significantly due to the increased use of digital services, reflecting a broader trend noted in studies of related health conditions. Moreover, discussions on public health governance suggest a critical need for democratic frameworks to support such digital transformations effectively.

## 1. Introduction

Digital health services encompass a broad spectrum of technologies and applications that leverage digital tools to enhance healthcare delivery, patient engagement, and health information management. These services range from telemedicine and mobile health applications to electronic health records and health information exchanges. They aim to improve the accessibility, efficiency, and quality of healthcare, thereby addressing disparities in service delivery. As highlighted in studies, the implementation of eHealth can facilitate better communication between providers and patients, while also allowing for effective data sharing and enhanced collaborative healthcare practices [1]. However, the rapid evolution of digital health services can be challenging, especially during public health crises, as it necessitates the integration of emergency management strategies with health delivery systems [2]. Thus, the assessment of digital health services requires a nuanced understanding of both their potential benefits and the barriers to their widespread adoption in challenging contexts.

In recent years, the emergence of digital health services has become increasingly significant, particularly in the context of global crises such as the COVID-19 pandemic. These services, which encompass telehealth, mobile health applications, and online patient portals, have transformed the way in which healthcare is delivered and consumed. Digital health services enhance remote access to care, benefiting both providers and patients during crises [3]. However, the effectiveness and user satisfaction of these services can vary substantially, influenced by factors such as technological literacy, accessibility, and the nature of healthcare needs. This literature review aims to explore the diverse perspectives of service providers and users regarding digital health services during crises, thereby illuminating the strengths and weaknesses of such interventions. Through this investigation, we seek to underscore the implications for future healthcare delivery in an increasingly digital landscape [4].

The significance of digital health in crisis situations cannot be overstated, particularly as it facilitates the continuity of care amidst unprecedented disruptions. In the wake of global emergencies, such as the COVID-19 pandemic, healthcare systems have increasingly turned to digital platforms to maintain access to medical services for patients. The rapid adoption of telehealth interventions has enabled providers to deliver care without the need for in-person visits, thereby reducing the risk of contagion while ensuring that patients receive essential medical attention [5]. This transition to online services has illuminated the necessity of flexible and resilient healthcare models that can adapt to fluctuating circumstances. As emphasized by recent studies, incorporating information and communication technologies within healthcare has proven vital in navigating crises, fostering not just immediate responses but also long-term improvements in health service delivery. Such developments necessitate an ongoing investigation into their implications for both providers and users [6].

A thorough literature review serves as a cornerstone in understanding the multifaceted role of digital health services, especially during crises. By systematically analyzing existing studies, this review aimed to elucidate how these services cater to the needs of both service providers and users amid unprecedented challenges. Such an examination reveals patterns in service delivery and the user experience, critical in tailoring interventions in crisis conditions. For instance, recent initiatives, such as the mental health co-responder program in Greeley and Evans, underscore the effectiveness of integrating specialized care into emergency response frameworks, thus avoiding unnecessary incarceration and fostering appropriate mental health care [7]. Moreover, enhancing information ecosystems is vital to empower communities during disruptions, facilitating their adaptation to change. Therefore, this literature review is not merely a summary but an analytical synthesis that highlights gaps and reveals future directions in digital health service research.

The primary research problem centers on understanding not only how specific digital healthcare services emerged as crucial solutions during this crisis but also how they can be sustained and optimized for future use in diverse contexts. In this light, the objectives of this research are to critically analyze existing digital healthcare solutions, assess their effectiveness in crisis response, and identify potential pathways for sustainable implementation as healthcare continues to evolve. Addressing these objectives not only contributes to the academic discourse surrounding digital health but also offers practical guidelines for healthcare providers, policymakers, and stakeholders engaged in future crisis preparedness planning. Overall, this review not only satisfies the research objectives but also addresses critical gaps identified in the healthcare delivery literature, ultimately fostering an ongoing dialog among stakeholders regarding the viability and sustainability of digital healthcare technologies in times of crisis.

## 2. Methods

In recent years, the integration of digital healthcare services has transformed the landscape of medical delivery, particularly in response to crises like the COVID-19 pandemic, which highlighted the urgent need for innovative solutions in healthcare delivery [8]. Amidst this evolution, the challenges surrounding the effective implementation and adoption of such services must be critically examined, as the prior literature indicates both the potential for success and the risk of widening disparities in healthcare access and equity. The primary research problem addressed in this review arises from the need to understand the complexities and implications of utilizing digital healthcare services in times of crisis, which include not only technological adaptation but also the responsiveness of healthcare systems to emerging needs [9]. Consequently, the objectives of this section include developing a methodological framework to systematically examine the existing literature, identify significant themes, and evaluate the effectiveness of digital innovations during crises. Furthermore, this section aims to synthesize insights regarding both the benefits and limitations of digital healthcare integration for a comprehensive understanding [10].

For this review, research was carried out on the Medline/PubMed, Scopus, and Google Scholar databases to locate articles published from 2015 to 2025, using the search terms digital health, digital technologies, management implications, and digital platforms. The inclusion criteria referred to studies that were directly related to the topic, available in the English language, and published in peer-reviewed scientific journals. The exclusion parameters were as follows: (a) articles not relevant to the topic as defined in the purpose of the review, (b) systematic reviews and meta-analyses, and (c) articles published in a language other than English.

## 3. Results

In crisis situations, digital health services play a pivotal role by enhancing access to care, streamlining communication among providers, and offering timely interventions. The increasing reliance on technology in healthcare delivery provides opportunities to address the treatment gap seen across various health services, particularly in mental health care, where barriers to access can hinder effective support [11]. Digital platforms allow for the provision of e-mental health resources, which can augment traditional methods and facilitate a continuum of care that aligns with patients’ needs. Additionally, insights from focus groups underscore that the attitudes of users towards digital health solutions are significantly influenced by their experiences with conventional healthcare systems [12,13]. These perspectives highlight a crucial tension between expectations for preventive strategies versus crisis management during service development [14]. Consequently, leveraging digital health services during crises not only improves immediate care but also reshapes long-term health service delivery models.

The historical context of digital health services is deeply intertwined with the evolution of technology and the shifting landscape of public health challenges. As governments worldwide grapple with crises such as political instability, economic downturns, and public health emergencies like the COVID-19 pandemic, the impetus for digital transformation within healthcare has intensified [15]. These crises have exposed systemic vulnerabilities, prompting rapid acceleration towards digital solutions to enhance healthcare accessibility and efficiency. Furthermore, the OECD emphasizes that modern governance must adapt to emerging threats, reinforcing democratic structures while promoting transparency and trust in public institutions [16]. As digital health services emerge as vital tools in managing these crises, they also serve to illustrate the broader challenges of governance and public engagement in the digital age, highlighting the crucial intersection between technology and democratic resilience [17]. This ongoing evolution underscores the need for the critical evaluation of digital health services within the context of contemporary societal challenges [18].

The adoption of digital health services has demonstrated marked acceleration during times of crisis, reflecting a need for efficient, accessible healthcare delivery. This shift is illustrated by the increased use of wireless handheld devices among healthcare professionals, which support seamless data collection and are particularly beneficial for patient care during emergencies [19]. Moreover, in the context of evolving customer demands, organizations have recognized that digital innovation is essential in optimizing healthcare services, thereby mitigating inefficiencies that arise during challenging times [20]. As a result, crises serve not only as catalysts for change but also highlight the urgent need for robust digital infrastructure that can withstand future disruptions, ultimately transforming the landscape of healthcare delivery [15].

### 3.1. Challenges Faced in the Deployment of Digital Health Services

The deployment of digital health services, particularly in times of crisis, faces a multitude of challenges that impede their effectiveness and accessibility. A salient issue is the integration of technology with existing healthcare systems, which often demonstrates significant incompatibility, thereby hindering seamless data exchange between platforms. Moreover, the reliance on advanced sensors and devices, while promising in improving healthcare delivery, presents barriers related to performance reliability and user interface compatibility, as noted in [21]. Additionally, there are non-technical challenges such as regulatory hurdles and ethical considerations that complicate the landscape of digital health. These challenges are further exacerbated by the need for robust digital literacy among both service providers and users, as disparities can lead to unequal access and diminishing trust in digital health solutions. Thus, addressing these multifaceted challenges is crucial for the successful implementation and sustainability of digital health services in a crisis context [22].

In exploring the perspectives of service providers within digital health services during crises, it is essential to acknowledge their unique insights and experiences. Many providers have witnessed firsthand the transformative impact of digital platforms in enhancing care delivery, especially under the constraints posed by situations such as the COVID-19 pandemic [23]. These professionals often highlight the necessity of effective training in emerging technologies to facilitate seamless integration into existing healthcare frameworks. The evolving landscape demands that providers adapt to new methods of governance and operational efficiency, thereby reinforcing their pivotal role in safeguarding patient rights while promoting health equity.

As service providers increasingly engage with digital health services, their experiences reveal both challenges and opportunities that are crucial for effective healthcare delivery [16]. Providers report that, while digital technologies can facilitate access to care, they also highlight significant barriers related to resource limitations, such as funding and workforce capacity, which impede their ability to offer comprehensive services. In particular, the need for collaborative models of care becomes apparent, as teamwork among health professionals enhances resource utilization and improves patient outcomes [11]. The intersection of ethnicity and care provision underscores the necessity of cultural competency among service providers. It is essential to recognize that disparities in healthcare access and quality can be exacerbated by communication barriers between providers and underserved populations [24]. Thus, while digital health services present novel avenues for care, the experiences of service providers emphasize the importance of addressing systemic issues to optimize health outcomes during crises [25].

In crisis situations, service providers encounter a myriad of barriers that hinder their ability to deliver effective digital health services. These challenges stem from both structural issues and technological limitations [13]. For instance, the lack of adequate information and communication technology (ICT) infrastructure can significantly impede service delivery, as evidenced by findings that highlight the shortages in IT physical infrastructure and unstable internet connections [26]. Additionally, service providers often grapple with a lack of experience and training in utilizing digital platforms, which can exacerbate feelings of frustration and anxiety during a crisis. The cited study also emphasizes the importance of understanding service providers’ perspectives on their information experiences, suggesting that historical and contextual factors play critical roles in shaping these challenges [27]. Addressing these barriers is essential in improving the efficacy of digital health services, particularly in times of widespread emergency.

Additionally, digital health platforms facilitate efficient communication between providers and patients, streamlining workflows and reducing administrative burdens [28]. As governments grapple with multiple crises and structural challenges, they are actively exploring digitalization as a means to bolster healthcare resilience and improve public health responses. This shift aligns with the broader goal of reinforcing democratic governance and public trust in institutional systems, emphasizing that effective digital health services not only support individual rights and freedoms but also enhance the capacity of systems to deliver sustained health benefits in the long term [29].

### 3.2. Digital Health Services from Users’ Perspectives

Understanding the perspectives of users is crucial in the development and implementation of digital health services, particularly during times of crisis. Users, such as cancer patients, emphasize the necessity of digital health technologies being user-friendly, effective in managing their conditions, and facilitating clear communication with healthcare providers [29]. This user-centric approach can greatly enhance patient engagement and satisfaction, leading to improved healthcare outcomes. Moreover, broader citizen attitudes towards digital health ecosystems indicate a demand for convenience in managing health services, supplemented by solid data protection measures [28]. As digital health solutions continue to evolve, integrating user feedback becomes imperative in ensuring that these technologies not only meet the practical needs of patients but also address their concerns regarding privacy and data security. Thus, recognizing user perspectives is fundamental in shaping effective digital health strategies during healthcare crises.

User experiences with digital health services during crises reveal both challenges and opportunities that significantly influence healthcare outcomes. As noted during the COVID-19 pandemic, there were substantial shifts in maternity care services, which affected both users and healthcare providers [12]. Users found that the transition to virtual care provided increased convenience and flexibility; however, the quality of care was often diminished due to concerns over digital exclusion and the inability to access in-person support networks [30]. Additionally, individuals from refugee and asylum seeker backgrounds highlighted communication and sociocultural factors that were critical in shaping their antenatal care experiences, often facing barriers that compromised their access to essential services. These findings underscore the importance of integrating user feedback into digital health strategies to enhance their accessibility and effectiveness, ultimately fostering a more inclusive and responsive healthcare system during times of crisis [31].

The accessibility and usability of digital health platforms have emerged as critical components in the effective delivery of healthcare services, particularly during times of crisis. As the integration of cloud computing into health information systems demonstrates, these technologies can significantly enhance accessibility while simultaneously reducing costs. This transition allows for the secure storage and sharing of electronic health records (EHR), facilitating continuity of care among authorized users [32]. However, the success of such systems relies on the quality of the data that they contain, as poor data quality can severely undermine the overall effectiveness of care delivery. Moreover, establishing trust in digital health platforms is paramount; users must feel confident that the information accessed is reliable and secure. Understanding how perceptions of trust shape user behavior is crucial in fostering engagement with these emerging digital services [33]. Thus, addressing both accessibility and usability remains essential for successful digital health implementation.

In the realm of digital health services, trust and privacy concerns emerge as pivotal factors influencing user engagement and satisfaction, particularly during crises. Users often express apprehension regarding the security of their personal health information, fearing that sensitive data may be breached or misused. This anxiety is compounded by the rapid implementation of digital mental health technologies (DMHTs), where the alignment of usability with therapeutic goals becomes paramount in fostering trust among users, as highlighted in the literature [34]. Furthermore, the effectiveness of digital tools like WardSonar underscores the importance of proactive measures in addressing safety and privacy, allowing patients to communicate their perceptions of ward safety in real time [35]. Consequently, it is essential for developers and providers of digital health services to cultivate a robust framework of privacy safeguards while enhancing transparency to build user trust amid ongoing health crises.

Reaffirming the central theme of equity, this review has highlighted the imperative to create collaborative approaches among stakeholders—including policymakers, healthcare providers, and community members—to ensure that technological advancements do not exacerbate existing health disparities. The literature has pointed to the importance of community involvement in the development of digital health strategies, affirming that ethical frameworks should incorporate diverse perspectives to foster trust and usability among all user populations [36,37].

The recognition of social determinants of health as fundamental components in the design and implementation of digital interventions emerges as a core theme, underscoring the need for context-specific policy responses that are sensitive to the unique challenges faced by various demographics [38,39].

However, despite the wealth of insights, the literature is not without limitations. Many studies have focused on the theoretical underpinnings of ethical frameworks but fall short in providing actionable guidelines that practitioners can implement [40]. Moreover, there is an observable scarcity of quantitative data that can substantiate claims regarding the impact of digital health technologies on various populations, emphasizing the need for further empirical research.

Demographic factors significantly shape user perspectives on digital health services, particularly in times of crisis. Age, gender, and socioeconomic status influence how individuals perceive and engage with these technologies. For instance, older adults often exhibit reluctance towards automated services due to a preference for human interaction, fearing that technology may compromise personalized care [41].

This hesitation is compounded by the notion that automation could diminish the role of human counselors, thereby impacting the quality of support received. Conversely, younger demographics may be more likely to embrace artificial intelligence and digital tools, viewing them as efficient means of accessing care, especially during crisis situations [42]. Furthermore, studies evaluating local health departments reveal that retention in care varies according to demographic characteristics, emphasizing the need for tailored interventions to engage diverse populations effectively. Understanding these variances is crucial in optimizing digital health service delivery during critical periods.

## 4. Discussion

The findings reveal that telemedicine and mobile health applications have effectively maintained continuity of care amidst widespread constraints on in-person visits, demonstrating a marked increase in usage and patient engagement during the pandemic [43]. Notably, studies indicate that these digital solutions helped to mitigate the impact of the viral outbreak on healthcare delivery by facilitating remote consultations and chronic disease management, as observed in various healthcare settings, which persisted even under the stress of a global crisis [44].

Comparatively, research conducted prior to the pandemic illustrated the slower uptake of these technologies, primarily due to regulatory barriers and limited infrastructure [9]. However, recent investigations suggest that the pandemic acted as a catalyst for change, prompting healthcare systems to adapt more swiftly to digital innovations, as seen in post-pandemic analyses [10]. The implications of these shifts are profound; they indicate the potential for the long-term integration of digital healthcare services into routine practice, transforming patient–provider interactions [45].

Moreover, the literature suggests a need for revised policy frameworks that address the technological and infrastructural gaps identified during the crisis [46]. Enabling widespread access to these services not only enhances patient outcomes but also aligns with sustainability goals by decreasing the carbon footprint associated with conventional healthcare delivery systems [47]. Furthermore, the emphasis on digital solutions raises important questions regarding data privacy and ethical considerations, which must be prioritized as the sector advances [48].

From a theoretical standpoint, this body of research underscores the necessity of interdisciplinary approaches to effectively address the complexities inherent in digital healthcare implementation [49]. Methodologically, future studies could benefit from longitudinal analyses that evaluate the ongoing effectiveness of these services beyond the initial crisis response phases [50]. Ultimately, these findings contribute to a growing discourse on how technology can be harnessed to build resilience in healthcare systems, offering actionable insights for practitioners and policymakers alike [51].

Thus, the digital transformation of healthcare not only responds to immediate crises but also sets the foundation for a more adaptive and innovative healthcare landscape moving forward [52]. Developing strategies that ensure equitable access and quality in digital services will be essential to fully realize the potential benefits [53]. This discourse positions digital healthcare not merely as a temporary measure but as a fundamental component of future healthcare strategies [54].

## 5. Conclusions

The findings clearly indicate that these digital services not only facilitate remote consultations but also enhance patient engagement, ultimately improving healthcare outcomes [55]. The implications of this research extend beyond the academic sphere, offering practical insights for healthcare providers and policymakers alike. By identifying the barriers to equitable access and the infrastructure challenges that were further accentuated during the COVID-19 pandemic, this investigation underscores the necessity of targeted policy interventions to bolster the adoption of digital solutions throughout healthcare systems [44].

There also exists a compelling need for interdisciplinary collaboration between healthcare providers, technologists, and policymakers to develop robust frameworks that ensure equitable access to these services [46]. Given the rapid advancement in digital technologies, ongoing research into emerging trends, such as artificial intelligence integration and the role of big data analytics in digital healthcare, would be invaluable. Ultimately, the findings of this review encourage a proactive approach to healthcare innovation, emphasizing the need for adaptable strategies that respond to the evolving landscape of healthcare delivery [48]. These insights suggest that ensuring equitable access to healthcare requires not just technological integration but also the provision of patient education and support to navigate these new systems. The implications of these findings extend beyond the academic discourse, as they highlight the need for policy enhancements that facilitate infrastructure development in underserved areas while addressing the disparities that digital healthcare can exacerbate when not equitably implemented [43]. Practically, healthcare organizations must consider integrating comprehensive training programs aimed at improving patients’ technological literacy as a critical component of healthcare delivery. Moreover, collaboration between healthcare providers and technology companies can foster innovations that cater to diverse patient needs and preferences [49]. Future research should focus on longitudinal studies to assess the long-term impacts of telehealth on patient outcomes across different demographics and geographic regions to build a robust evidence base for ongoing improvements in digital health delivery. Further investigations into the efficacy of hybrid models that combine traditional in-person care with telehealth services are also recommended to provide insights for the optimization of patient engagement and satisfaction [36]. Additionally, studies examining the integration of mental health services into telehealth frameworks could provide significant benefits for populations that experience heightened barriers to in-person visits, such as those with mental illnesses [55]. Consequently, it is essential for ongoing research to consider the ethical implications and the need for policies that protect patient privacy while fostering innovation in digital health initiatives.

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
