# Peer review of "Digital Healthcare Innovative Services in Times of Crisis: A Literature Review"

_healthcare, 2025, doi:10.3390/healthcare13080889_

Round 1

Reviewer 1 Report

Comments and Suggestions for Authors

The manuscript titled "Digital Health Services in Times of Crisis: A Literature Review" provides a comprehensive exploration of the role, challenges, and implications of digital health services in crisis scenarios. The authors successfully compile an extensive review of existing literature, highlighting critical themes such as accessibility, workforce training, cost implications, and governance structures. While the paper presents valuable insights, several areas could be improved in terms of structure, coherence, methodological rigor, and clarity.

Areas for Improvement

  • Structure and Organization: The manuscript covers multiple critical areas, but the flow of information is somewhat disjointed. The Introduction is extensive and somewhat repetitive, making it difficult for the reader to grasp the core objectives early on. Some sections (e.g., Historical Context of Digital Health Services) feel disconnected from the rest of the discussion.
    • Suggestion: Introduce a clearer framework that outlines key themes and sections systematically.

  • Methodological Clarity: The paper lacks a well-defined methodology section. It is unclear whether this is a systematic or scoping review. Inclusion and exclusion criteria for the reviewed literature are not explicitly mentioned.
    • Suggestion: Add a section detailing the search strategy, databases used, criteria for selection, and any limitations in data collection.

  • Depth of Analysis: The paper provides an overview of digital health services, there is limited critical engagement with the studies cited. Some claims (e.g., “digital health increases disparities among vulnerable populations”) need stronger evidence and references. The discussion on financial implications lacks data-driven insights (e.g., specific cost analyses, policy comparisons).
    • Suggestion: Deepen the discussion by critically analyzing study findings rather than summarizing them.

  • Clarity and Conciseness: Several paragraphs repeat similar ideas, leading to redundancy. Certain sentences are overly complex, making it harder for readers to follow the argument.
    • Example: "By facilitating remote access to medical consultations and resources, digital health services offer critical support to both providers and patients during challenging times." This could be simplified as: "Digital health services enhance remote access to care, benefiting both providers and patients during crises."
    • Suggestion: Revise for conciseness and readability.

  • Ethical and Socioeconomic Considerations: The discussion of health equity and social determinants of health is somewhat superficial. While digital health services are noted to exacerbate inequities, policy recommendations to address these gaps are missing.
    • Suggestion: Expand on potential policy interventions and ethical frameworks that could guide equitable digital health adoption.

  • Conclusion and Future Directions: The conclusion effectively summarizes the main findings but lacks specific, actionable recommendations for policymakers and practitioners. The section on Future Research Directions is vague and should be more targeted.
    • Suggestion:  Provide clear research priorities (e.g., evaluating AI in telehealth, developing digital literacy programs). Recommend specific policy actions (e.g., subsidies for low-income populations, workforce training programs).

Author Response

Comment 1: Add a section detailing the search strategy, databases used, criteria for selection, and any limitations in data collection.

Response: We added the method section (highlighted text).

Comment 2: Deepen the discussion by critically analyzing study findings rather than summarizing them

Response: We added the discussion section (highlighted text).

Comment 3: Several paragraphs repeat similar ideas, leading to redundancy. Certain sentences are overly complex, making it harder for readers to follow the argument.

Example: "By facilitating remote access to medical consultations and resources, digital health services offer critical support to both providers and patients during challenging times." This could be simplified as: "Digital health services enhance remote access to care, benefiting both providers and patients during crises."

Response: We have omitted the paragraphs with similar ideas. We have done the change in the phrase (highlighted text).

Comment 4: Expand on potential policy interventions and ethical frameworks that could guide equitable digital health adoption.

Response: We expanded the text. (highlighted text, p.6)

Comment 5: Provide clear research priorities (e.g., evaluating AI in telehealth, developing digital literacy programs). Recommend specific policy actions (e.g., subsidies for low-income populations, workforce training programs).

 Response:  We changed the conclusion section (highlighted text, p.8)

Reviewer 2 Report

Comments and Suggestions for Authors

The authors present a review-type study entitled: "Digital Health services in times of crisis; A Literature Review"

The literature review is important given the solutions used during crisis medical care. To improve the quality of the manuscript, the authors should explain the following:

1. The manuscript's writing style must be scientific and supported by a greater number of references. For example: 1.3 Types of Digital Health Services Utilized During Crises only has three references [8-11]. Another example: Historical Context of Digital Health Services has one reference [again, reference 10].

2. To conduct a "Literature Review," it is necessary to present the selected methodology. What were the search parameters? The exclusion parameters? etc.

3. The title is very promising, and the content should be more comprehensive. The sections of the manuscript are very basic and offer little contribution to the state of the art.

4. The abstract should include: context, problem, objective, method, results, and conclusions.

5. The Conclusions section should be expanded to include a discussion of more than just six references.

6. Authors should follow the MDPI journal and publisher's format. This should be reviewed in detail.

7. References 12 from 2005 and 25 from 2002 should be updated.

Author Response

number of references. For example: 1.3 Types of Digital Health Services Utilized During Crises only has three references [8-11]. Another example: Historical Context of Digital Health Services has one reference [again, reference 10].

Response: We have added more references

Comment 2:  To conduct a "Literature Review," it is necessary to present the selected methodology. What were the search parameters? The exclusion parameters? etc.

Response: We have added the methods section (highlighted text)

Comment 3: The title is very promising, and the content should be more comprehensive. The sections of the manuscript are very basic and offer little contribution to the state of the art.

Response: We have rewritten many sections of the manuscript (highlighted text)

Comment 4: The abstract should include: context, problem, objective, method, results, and conclusions.

Response: We have rewritten the abstract. (highlighted text)

Comment 5: The Conclusions section should be expanded to include a discussion of more than just six references.

Response: We have rewritten the conclusion adding references. (highlighted text)

Comment 6: Authors should follow the MDPI journal and publisher's format. This should be reviewed in detail.

Response: We have review the manuscript. (highlighted text)

Comment 7: References 12 from 2005 and 25 from 2002 should be updated.

Response: We have updated the references

Reviewer 3 Report

Comments and Suggestions for Authors

Digital Health Services in Times of Crisis: A Literature Review

The topic is highly current, particularly in the post-COVID-19 context.
The paper includes recent and relevant sources, especially concerning technology, digital health, and emergency responses.

The title could be improved to better reflect the scope and focus of the review.
The sentence structure in the abstract is confusing. It should be rewritten using shorter and more concise sentences.

The objectives of the review are not clearly stated or easily identifiable.
Redundancies: Some paragraphs throughout the article contain repeated ideas. There is redundancy in certain sections (e.g., 1.1 and 1.4), and excessive repetition of references such as [8], [10], [15], [23], [24].
Language and referencing: From a linguistic perspective, the English requires improvement. Commas are often missing, and some references are not clearly or correctly presented. The citation format should also be revised

Comments on the Quality of English Language

The title could be improved to better reflect the scope and focus of the review.
The sentence structure in the abstract is confusing. It should be rewritten using shorter and more concise sentences.

Author Response

Comment 1: The title could be improved to better reflect the scope and focus of the review.

Response: We changed the title. (highlighted text)

Comment 2: The sentence structure in the abstract is confusing. It should be rewritten using shorter and more concise sentences.

Response: We rewrote the abstract. (highlighted text)

Comment 3: The objectives of the review are not clearly stated or easily identifiable.

Response: We revised the objectives (highlighted text)

Comment 4: Redundancies: Some paragraphs throughout the article contain repeated ideas. There is redundancy in certain sections (e.g., 1.1 and 1.4), and excessive repetition of references such as [8], [10], [15], [23], [24].

Response: We deleted the repeated ideas. (highlighted text)

Comment 5: Language and referencing: From a linguistic perspective, the English requires improvement. Commas are often missing, and some references are not clearly or correctly presented.

Response: We revised the text. (highlighted text)

 Comment 6: The citation format should also be revised

Response: We revised the citation format

Round 2

Reviewer 2 Report

Comments and Suggestions for Authors

The authors have reviewed and corrected all of the reviewer's comments.

I only believe they should correct the title to: Digital Healthcare Innovative Services in Times of Crisis: A Literature Review

Author Response

Comment 1: I only believe they should correct the title to: Digital Healthcare Innovative Services in Times of Crisis: A Literature Review

Response: We changed the title (highlighted text).

Reviewer 3 Report

Comments and Suggestions for Authors According to reference 40, the issue of health equity stems from equity in access to technology. This is neither clarified nor addressed in the article. This should be rectified. The objectives need to be more clearly defined. The article must fully reflect its intended purpose. The aim must be explicit. For this reason, I propose minor revisions.

Author Response

Comment 1: According to reference 40, the issue of health equity stems from equity in access to technology. This is neither clarified nor addressed in the article. This should be rectified. The objectives need to be more clearly defined. The article must fully reflect its intended purpose. The aim must be explicit. For this reason, I propose minor revisions.

Response: We omitted the reference. We clarified the objectives in abstract and in the text (highlighted text, p. 3).